# Effect of *Cladosporium cladosporioides* on the Composition of Mycoflora and the Quality Parameters of Table Eggs during Storage

Pavlina Jevinová, Monika Pipová *, Ivana Regecová, Soňa Demjanová, Boris Semjon, Slavomír Marcinčák, Jozef Nagy and Ivona Kožárová

Department of Food Hygiene, Technology and Safety, University of Veterinary Medicine and Pharmacy, Komenského 73, 041 81 Košice, Slovakia; pavlina.jevinova@uvlf.sk (P.J.); ivana.regecova@uvlf.sk (I.R.); sona.demjanova@student.uvlf.sk (S.D.); boris.semjon@uvlf.sk (B.S.); slavomir.marcincak@uvlf.sk (S.M.); jozef.nagy@uvlf.sk (J.N.); ivona.kozarova@uvlf.sk (I.K.)
* Correspondence: monika.pipova@uvlf.sk; Tel.: +421-915984562

**Abstract:** The eggshells of 120 experimental one-day-old table eggs were contaminated with the spore suspension of *Cladosporium cladosporioides*, divided into three groups (A–C) and stored at three different temperatures (3 °C, 11 °C and 20 °C) for 28 days. Visible growth of molds on/in experimental eggs was not observed within the entire storage period. No significant differences in the numbers of molds were found between particular groups of eggs. However, the composition of egg mycoflora was greatly influenced by storage conditions. Three mold genera were identified using the PCR method. The highest mold numbers were determined on Day 14 (Groups A and C) and Day 21 (Group B) when the maximum relative humidity and dew point temperature were recorded. On the same days, the dominance of *Penicillium* spp. and the minimum eggshell firmness were observed. Noticeable changes in egg quality were observed in eggs stored at 20 °C, and most of these eggs were downgraded at the end of storage period. The growth ability differed significantly among three mold genera. *Penicillium* spp. and *Fusarium* spp. showed better growth intensity at increased values (0.91–0.94) of water activity ($a_w$) indicating a possible risk associated with the occurrence of mycotoxins in the egg contents.

**Keywords:** egg quality; mold; *Cladosporium*; *Penicillium*; *Fusarium*

## 1. Introduction

*Cladosporium* (*C.*) spp. belong to the most common molds in indoor and outdoor air, as well as in materials such as soil, plants, textiles, plastics and foodstuffs [1–6]. Small, dry and heavily pigmented conidia provide fungus with very effective protection against ultraviolet radiation and enable a long-term persistence in environments including various substrates [7]. According to the most recent information, *Cladosporium* genus includes 218 species classified into three species complexes—*C. herbarum*, *C. sphaerospermum* and *C. cladosporioides* [4,7–9]. Both *C. sphaerospermum* and *C. cladosporioides* have already been isolated from foodstuffs, indoor air and materials from households [1,7]. The presence of *C. cladosporioides* was reported in fresh vegetables, wheat, flour, barley, rice, dried fish, cheeses, sea salt, as well as in table eggs [10–12].

The risk of egg contamination is related to the penetration of microorganisms from the outside environment through the natural protective egg barriers into the egg contents [13]. With the exception of endogenous contamination, the contents of freshly laid eggs usually do not show any presence of microorganisms. The eggshell becomes contaminated with faeces in the cloaca at the moment of oviposition. Further contamination of the shell occurs immediately after laying and results from direct contact with contaminated surfaces, floor litter or nesting materials [14]. As soon as the eggshell has been exposed to the

environment, it becomes heavily contaminated with various microorganisms, including the spores of molds that are therefore permanently present on the eggshell surface. Spore numbers depend on the level of contamination on egg farms and establishments for further handling and grading of eggs. Salem et al. [15] reported significantly higher numbers of spores on the surface of visibly contaminated eggshells (5.6 log CFU/g) compared with those that were uncontaminated (4.4 log CFU/g). Although the spectrum of mold species varies considerably, the most frequent mold genera found on the shell surface include *Aspergillus* spp., *Penicillium* spp., *Cladosporium* spp., *Rhizopus* spp. and *Mucor* spp. [16–18].

Generally, foods of animal origin (including table eggs) have a limited shelf life at low storage temperatures. When stored under suitable conditions (i.e., at low temperature and high relative humidity), germination of spores and growth of contaminating mycoflora are observed. Hyphae of molds are able to penetrate through the shell pores into the egg contents. This process is accompanied by the destruction of both shell membranes, thus enabling further bacterial growth. At the final stage of mold decomposition, coagulation or gelatinization of the egg albumen is observed and an unpleasant 'moldy' smell is developed, which persists even after heat processing. In such a way, table eggs become devalued and inappropriate for human consumption [19].

Due to its psychrophilic nature and growth ability at low temperatures (up to $-5\,^\circ$C), *C. cladosporioides* is capable of causing undesirable changes in table eggs during storage. This fungal species is reported to grow at a temperature of $25\,^\circ$C and water activity ($a_w$) below 0.86 [20,21].

The aim of this study was to reveal the effect of storage conditions on the growth of molds and the related changes in the quality of table eggs previously contaminated with spores of *C. cladosporioides* during the shelf-life period.

## 2. Materials and Methods

### 2.1. Materials

The experiment was performed with 120 one-day-old table eggs. Eggshells were contaminated by the short immersion of experimental eggs into the spore suspension of *C. cladosporioides* CCM F-348 (Czech Collection of Microorganisms, Brno, Czech Republic) standardized to the density of a McFarland 0.8 turbidity standard, approximately corresponding to 6.4 log CFU/mL. After drying the shells in air, table eggs were divided into three groups (40 eggs per group), placed into cardboard boxes and stored under different conditions commonly used for the storage of eggs in households.

Group A eggs were stored in the refrigerator with forced-air convection (Liebherr, Ochsenhausen, Germany) at an average temperature of $11\,^\circ$C. Group B eggs were kept in the refrigerator without forced-air convection (AEG, Berlin, Germany) at $3\,^\circ$C, and Group C eggs were stored at laboratory temperature ($20\,^\circ$C). Storage temperature, relative air humidity and dew point temperature were monitored permanently within the entire storage period using the Temperature Humidity Transmitter TFA 30.3180.IT TTH (ELSO Philips Service s.r.o., Trenčín, Slovakia). Shell surface, egg albumen and egg yolk were sampled on Days 7, 14, 21 and 28 of the experiment for enumeration of molds and evaluation of selected qualitative parameters. Shell firmness was determined with the help of the Egg Force Reader (Orka Food Technology Ltd., Herzliya, Israel), and the Egg Analyzer$^{\text{TM}}$ (Orka Food Technology Ltd., Herzliya, Israel) was used for automatic measurements of the egg weight, the yolk color, automatic calculation of Haugh units and assessment egg quality grades.

### 2.2. Enumeration of Molds

Molds were removed from the eggshell surface as described by Cupáková et al. [22]. Five eggs from each experimental group were tested every week. Each individual egg was transferred aseptically into a sterile plastic bag, and sterile peptone water (0.1%) in a volume of 100 mL had been added. The sample was then shaken for 15 min using the Orbi-ShakerTMJR (BioTech s.r.o., Bratislava, Slovakia).

Shell surface was decontaminated with alcohol and then cut with the help of a sterile scalpel. Egg yolk and egg albumen were captured separately into sterile containers. A test sample of the egg albumen/yolk in the amount of 10 g was homogenized with 90 mL of sterile peptone water (0.1%) for 2.5 min using the peristaltic BagMixer (Interscience, Nom la Brétèche, France). Decimal dilutions were prepared in accordance with STN EN ISO 6887-1 [23]. A volume of 0.1 mL of each appropriate decimal dilution was spread on the surface of Dichloran Rose Bengal Chloramphenicol agar medium (DRBC, Oxoid, Hampshire, UK). Colonies of molds were enumerated after incubation of the inoculated plates at 25 °C for 5 days [24].

### 2.3. Identification of Mold Genera

Colonies of molds were first identified visually on the surface of Czapek Yeast Extract Agar (CYEA), Yeast Extract Sucrose agar medium (YES), Sabouraud Dextrose Agar (SDA) and Potato Dextrose Agar (PDA, Oxoid, Hampshire, UK) within a 7-day incubation period. The following parameters were evaluated: growth rate; size, margin, morphology and radial ornamentation of the colony; color, structure and height of the aerial mycelium; and production of soluble pigments and exudates. Microscopic patterns of mold isolates were observed on slides using the lactophenol cotton blue staining procedure. The typical structure of hyphae, conidial chains, branching patterns, and sexual and asexual reproduction structures were identified using light microscopy [8,25,26].

### 2.4. Genus Confirmation with the Help of PCR Method

Due to the typical cell wall structure, both pre-isolation and isolation steps were used to obtain DNA from mold isolates. Mycelium in a quantity of 10–50 mg was removed from the surface of CYEA using a sterile scalpel and transferred into an Eppendorf tube with 0.2 mL of sterile zircon and glass beads (1:1 ratio). Proteinase K (Macherey-Nagel GmbH & Co., Dueren, Germany) in a volume of 10 μL was added to the sample, and the Eppendorf was incubated at 37 °C for 30 min. After incubation, 800 μL of the lysing solution FG1 (OMEGA Bio-tek, Inc., Norcross, GA, USA) was added to the sample, and the Eppendorf was incubated for another 10 min at 65 °C in ultrasound waves of 500 Hz. Subsequently, isolation of DNA was performed according to the instructions of commercially available E.Z.N.A.® Fungal DNA Mini Kit (OMEGA Bio-tek, Inc., Norcross, GA, USA). The purity and concentration of DNA obtained was checked with the help of BioSpec Nanometer Spectrophotometer (Shimadzu, Kjóto, Japan).

The forward primer Pen 1–F (5′-AAATATAAATTATTTAAAACTTTC-3′) and the reverse primer Pen 2–R (5′-CTGGATAAAAATTTGGGTTG-3′) were designed based on the Internal Transcribed Spacer (ITS) region and 5.8S rRNA sequences from *Penicillium* spp. available in the GenBank-EMBL [27]. Another two primers were designed to amplify fragments within the ITS regions of *Fusarium* spp. and then synthesized commercially by Amplia s.r.o. (Bratislava, Slovakia). The initial tests for specificity have revealed that the primer pair ITS-Fu-f (5′-CAACTC CCAAACCCCTGTGA-3′) and ITS-Fu-r (5′-GCGACGATTACCAGTAACGA-3′) is highly specific for *Fusarium* genus [28].

Mitochondrial small subunit rRNA of *Cladosporium* spp. was amplified using the universal fungal mitochondrial primers MS1 (5′-CAGCAGTCAAGAATATTAGTCAATG-3′) and MS2 (5′-GCGGATTATCGAATTAAATAAC-3′). Two specific primers, Clado-PF (5′-TACTCCAATGGTTCTAATATTTTCCTCTC-3′) and Clado-PR (5′-GGGTACCTAGACAGTA TTTCTAGCCT-3′), were designed for multiplex PCR assay and synthesized by Amplia s.r.o. (Bratislava, Slovakia). The expected amplicon size for the primer pair Clado-PF/R was 87 bp [29].

The amplification was performed in the Thermal Cycler TC-512 (Techne, Stadfforshire, UK) using the PCR mixture containing 1–10 μL of template DNA, 0.5 μL of each primer (concentration 10 pmol.μL$^{-1}$) and 4.0 μL of HOT Firepol® Blend Master Mix (Solis BioDyne, Tartu, Estonia) in a volume of 20 μL. The same amplification conditions were used for each of the primer sets: an initial denaturation of 12 min at 95 °C was necessary for activating

HOT Firepol polymerase, followed by 30 cycles of denaturation of 20 s at 95 °C, annealing of 60 s at different temperatures for each particular primer pair (46 °C for Pen1/Pen2; 55 °C for ITS-Fu-f/ITS-Fu-r; 51 °C for MS1/MS2 and Clado-PF/Clado-PR), elongation of 2 min at 72 °C, and final extension of 10 min at 72 °C. The final amplification temperature was 6 °C.

To determine the minimum amount of fungal DNA detectable by the established PCR assays, variable quantities of mold genomic DNA ranging from 10 to 100 ng were used as the DNA template. All the PCR products were size fractionated in agarose gels (1.5%), stained with the GelRed™Nucleic Acid gel stain (Biotium, Inc., Fremont, CA, USA) and visualized by the UV transilluminator Mini Bis Pro® (DNR Bio-Imaging Systems Ltd., Neve Yamin, Izrael). Sequences obtained from the studied mold isolates were submitted to the GenBank-EMBL database and searched for homology using the Basic Local Alignment Search Tool (BLAST) program (NCBI software package).

Data generated in this study are included in this article. Sequence data that confirm mold genera identified in this study can be found in the GenBank under the following accession numbers: FJ362555.1, FN386269.1 and MH517445.1.

### 2.5. Characteristics of Growth and Interactions of Selected Mold Species

Spore germination and growth intensity of *C. cladosporioides* CCM F-348, *C. herbarum*, *P. chrysogenum* CCM F-362, *P. crustosum* CCM F-8322, *P. griseofulvum* CCM F-8006, *P. glabrum* CCM F-310 and *F. graminearum* CCM F-683 obtained from the Czech Collection of Microorganisms (CCM, Brno, Czech Republic) were tested on both DRBC and SDA. Culture media were prepared in accordance with the manufacturer's instructions. As soon as they became solid, water activity at 25 °C was measured using the LabMASTER-a$_w$ (Novasina, Farmingdale, NY, USA). Fungal spore suspensions were adjusted to a McFarland 0.8 turbidity standard corresponding with a density of 6.4 log CFU/mL. After that, 10 μL aliquots of individual spore suspensions were applied to 6 mm sterile disks (BBL™, Los Angeles, CA, USA) placed on the surface of both DRBC and SDA agar plates. To test the spore germination and growth intensity, a single disk with spore suspension was placed to the center of a small size Petri dish (Ø 60 mm). To study fungal interactions, one disk was placed at the center of a medium size Petri dish (Ø 90 mm) and another four disks surrounded the central one at a distance of 1 cm. Fungal spore suspension was applied on the disks in various combinations in order to observe mutual interactions. After incubation of the inoculated plates at 25 °C for 120 h, both the spore germination and growth of hyphae were evaluated using the optical microscope (Lupa Hama, Monheim, Germany) at 10× magnification. As soon as visible colonies appeared on the culture media, the diameters were measured in 24-h intervals. The evaluation of mutual antifungal activity between *C. cladosporioides*, *Penicillium* and *Fusarium* species was based on the overall ability of a particular fungal strain to inhibit the radial growth of competitive mold genera.

### 2.6. Statistical Analysis

Pearson's correlation coefficients, two-way analysis of variance (ANOVA) and Tukey test for multiple comparison of means with a confidence interval set at 95% were conducted with R—statistics software. The effect of different storage conditions and the effect of the storage period were set as the main factors. Multiple factor analysis (MFA) was conducted in R—statistics software [30] with the "FactoMineR" [31] and "Factoextra" package [32] according to Semjon et al. [33].

## 3. Results

### 3.1. Effect of Storage Conditions on the Growth of C. cladosporioides and Competetive Fungal Species

In order to maintain stable storage conditions, three parameters (air temperature, relative humidity and dew point temperature) were permanently monitored during the entire storage period in this study. As seen in Table 1, storage parameters were significantly

different for each of three experimental groups of table eggs ($p < 0.001$). During storage, significant differences have been observed among storage temperatures in Group C ($p < 0.01$) as well as among the values of both relative humidity and dew point temperature in Group B ($p < 0.05$). However, the highest values of both the relative humidity and the dew point temperature were recorded on Day 14 (Groups A and C) and Day 21 (Group B).

Results of the quantitative mycological examination are shown in Table 2. No significant effect of storage conditions on the numbers of molds on the shell surface or in the egg contents was confirmed in Groups A and B. Conversely, storage period significantly influenced the number of molds on the shell surface of Group C eggs ($p < 0.01$). As seen in Table 2, the average mold numbers on/in the eggs remained practically unchanged across the entire storage period and did not show any statistical significance.

**Table 1.** Storage conditions for three experimental groups of table eggs (mean $\pm$ SD).

| Parameter | Group | Storage Period [Days] | | | | ANOVA [*p* Values] | |
| | | 7 | 14 | 21 | 28 | Impact of Storage Conditions | Impact of Storage Period |
|---|---|---|---|---|---|---|---|
| Temperature [° C] | A | 11.50 ± 0.30 [a2] | 11.44 ± 0.14 [a2] | 11.34 ± 0.15 [a2] | 11.46 ± 0.27 [a2] | | |
| | B | 2.27 ± 0.55 [a3] | 3.19 ± 0.90 [a3] | 4.13 ± 1.55 [a3] | 3.72 ± 0.77 [a3] | $p < 0.001$ | $p > 0.05$ |
| | C | 20.19 ± 0.78 [b1] | 22.09 ± 0.13 [a1] | 20.00 ± 0.86 [b1] | 20.12 ± 0.36 [b1] | | |
| Relative humidity [%] | A | 78.75 ± 4.38 [a1] | 80.98 ± 1.41 [a1] | 80.02 ± 2.53 [a1] | 76.25 ± 1.09 [a1] | | |
| | B | 54.00 ± 5.00 [b2] | 54.30 ± 4.39 [ab2] | 64.00 ± 2.62 [a1] | 58.74 ± 2.04 [ab2] | $p < 0.001$ | $p > 0.05$ |
| | C | 45.44 ± 2.87 [a2] | 45.47 ± 1.49 [a3] | 41.72 ± 11.99 [a2] | 40.75 ± 3.42 [a3] | | |
| Dew point [° C] | A | 7.93 ± 0.54 [a1] | 8.28 ± 0.19 [a1] | 8.00 ± 0.37 [a1] | 7.38 ± 0.31 [a1] | | |
| | B | −6.43 ± 0.32 [b2] | −4.83 ± 1.27 [ab2] | −2.07 ± 1.63 [a2] | −3.68 ± 1.20 [ab2] | $p < 0.001$ | $p > 0.05$ |
| | C | 8.02 ± 1.59 [a1] | 9.76 ± 0.40 [a1] | 6.26 ± 3.19 [a1] | 6.36 ± 1.56 [a1] | | |

Different superscripts in the same row ([a, b]) or column ([1–3]) indicate that mean values differ significantly (Tukey's test, $p < 0.05$).

**Table 2.** Numbers of molds in experimental groups of table eggs during the storage period (mean $\pm$ SD).

| Parameter | Group | Storage Period [Days] | | | | ANOVA [*p* Values] | |
| | | 7 | 14 | 21 | 28 | Impact of Storage Conditions | Impact of Storage Period |
|---|---|---|---|---|---|---|---|
| Eggshell [log CFU/egg] | A | 4.37 ± 0.31 [a] | 4.61 ± 0.12 [a] | 4.10 ± 0.17 [a] | 4.23 ± 0.72 [a] | | |
| | B | 4.29 ± 0.03 [a] | 4.67 ± 0.06 [a] | 4.75 ± 0.43 [a] | 4.28 ± 0.10 [a] | $p < 0.001$ | $p > 0.05$ |
| | C | 4.29 ± 0.17 [ab] | 4.78 ± 0.27 [a] | 4.42 ± 0.15 [ab] | 3.98 ± 0.32 [b] | | |
| Egg albumen [log CFU/g] | A | 0.00 ± 0.00 [a] | 2.00 ± 1.73 [a] | 0.67 ± 1.15 [a] | 0.87 ± 1.50 [a] | | |
| | B | 0.00 ± 0.00 [a] | 0.67 ± 1.15 [a] | 2.19 ± 1.90 [a] | 0.67 ± 1.15 [a] | $p < 0.001$ | $p > 0.05$ |
| | C | 0.00 ± 0.00 [a] | 1.77 ± 1.53 [a] | 0.67 ± 1.15 [a] | 0.00 ± 0.00 [a] | | |
| Egg yolk [log CFU/g] | A | 0.00 ± 0.00 [a] | 1.97 ± 1.71 [a] | 0.00 ± 0.00 [a] | 0.00 ± 0.00 [a] | | |
| | B | 0.00 ± 0.00 [a] | 0.67 ± 1.15 [a] | 2.18 ± 1.89 [a] | 0.67 ± 1.15 [a] | $p < 0.001$ | $p > 0.05$ |
| | C | 0.00 ± 0.00 [a] | 1.91 ± 1.66 [a] | 0.00 ± 0.00 [a] | 0.67 ± 1.15 [a] | | |

Different superscripts in the same row ([a, b]) or column ([1–3]) indicate that mean values differ significantly (Tukey's test, $p < 0.05$).

The composition of mycoflora changed noticeably during egg storage. Apart from *C. cladosporioides*, the fungal species used for initial eggshell contamination, another two mold genera (*Penicillium* spp. and *Fusarium* spp.) as well as yeasts have also been isolated from experimental eggs (Table 3 and Figure 1).

**Table 3.** Proportion of mold genera on the eggshell of experimental eggs stored at 11 °C (Group A), 3 °C (Group B) and 20 °C (Group C).

| Day | Group A | Group B | Group C |
| --- | --- | --- | --- |
| 1 | *C. cladosporioides* (100%) | *C. cladosporioides* (100%) | *C. cladosporioides* (100%) |
| 7 | *C. cladosporioides* (94%) *Penicillium* spp. (6%) | *C. cladosporioides* (100%) | *C. cladosporioides* (100%) |
| 14 | *C. cladosporioides* (9%) *Penicillium* spp. (91%) | *C. cladosporioides* (32%) *Penicillium* spp. (68%) | *C. cladosporioides* (20%) *Penicillium* spp. (80%) |
| 21 | *C. cladosporioides* (34%) *Penicillium* spp. (33%) *Fusarium* spp. (33%) | *C. cladosporioides* (12%) *Penicillium* spp. (88%) | *C. cladosporioides* (57%) *Penicillium* spp. (29%) Yeasts (14%) |
| 28 | *C. cladosporioides* (99%) *Penicillium* spp. (1%) | *C. cladosporioides* (100%) | *C. cladosporioides* (100%) |

*C.—Cladosporium.*

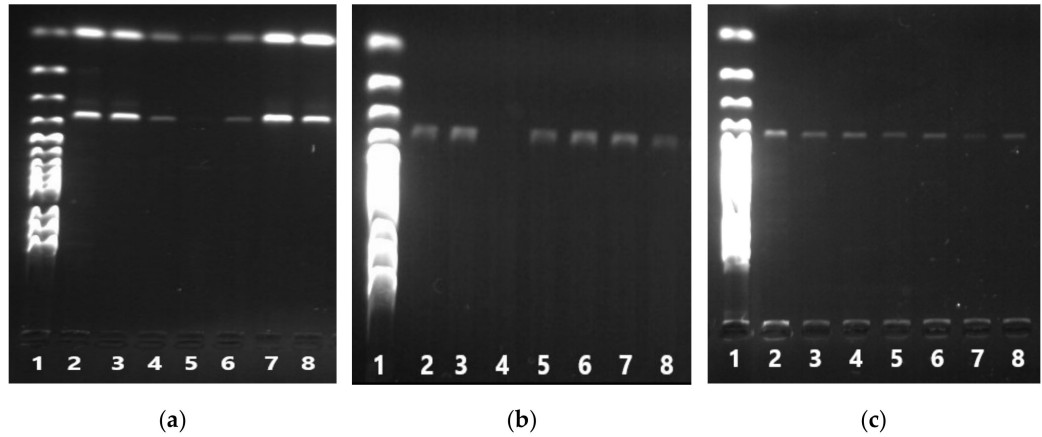

|  (**a**) | (**b**) | (**c**) |

**Figure 1.** Detection of individual mold genera by PCR: (**a**) *Cladosporium* spp. (87 bp; internal control 370 bp) Line 1: 100 bp ladder standard; Line 2: *Cladosporium cladosporioides* CCM F-348; Lines 3 to 8: isolates of *Cladosporium* spp.; (**b**) *Penicillium* spp. (336 bp) Line 1: 100 bp ladder standard; Line 2: *Penicillium chrysogenum* CCM F-362; Lines 3,5,6,7, and 8: isolates of *Penicillium* spp.; Line 4: non- *Penicillium* spp isolate; (**c**) *Fusarium* spp. (410 bp) Line 1: 100 bp ladder standard; Line 2: *Fusarium graminearun CCM* F-683; Lines 3 to 8: isolates of *Fusarium* spp.

*3.2. Growth Characteristics of Cladosporium, Penicillium and Fusarium Species*

As can be seen in Table 4, the growth of *C. cladosporioides* and *C. herbarum* on both DRBC and SDA was confirmed 24–72 h later than that of the *Penicillium* and *Fusarium* species used in this study. In addition, all *Penicillium* and *Fusarium* strains showed less intense growth with small colony diameters.

The interactions between the *C. cladosporioides*, *Penicillium* and *Fusarium* species were evaluated after a 120-h cultivation on both DRBC and SDA. The antifungal activity of *C. cladoporioides* was confirmed against *P. chrysogenum*, *P. crustosum* and *P. griseofulvum*. Moreover, mutual inhibition was observed among the four *Penicillium* species tested. On the other hand, the growth of *Fusarium graminearum* was not inhibited by any of the *Penicillium* and *Cladosporium* species tesed in this study (Figure 2).

**Table 4.** Colony diameters (Ø) of *Cladosporium*, *Penicillium* and *Fusarium* species grown on Dichloran Rose Bengal Chloramphenicol agar (DRBC) and Sabouraud Dextrose Agar (SDA) (mean ± SD).

| Fungal Species | Incubation Period (h) | | | | |
|---|---|---|---|---|---|
| | 24 | 48 | 72 | 96 | 120 |
| | DRBC (a$_w$ = 0.91) Ø (mm) | | | | |
| *C. cladosporioides* CCM F-348 | neg | pos$^+$ | 7.00 ± 0.00 | 9.67 ± 0.58 | 12.67 ± 0.58 |
| *C. herbarum* CCM F-455 | neg | neg | pos$^+$ | 8.33 ± 0.58 | 11.00 ± 0.00 |
| *P. chrysogenum* CCM F-362 | pos$^+$ | 8.00 ± 0.00 | 11.67 ± 0.58 | 14.33 ± 0.58 | 18.00 ± 0.00 |
| *P. crustosum* CCM F-8322 | pos$^+$ | 10.00 ± 0.00 | 12.00 ± 0.58 | 16.67 ± 0.58 | 19.67 ± 0.58 |
| *P. griseofulvum* CCM F-8006 | pos$^+$ | 8.33 ± 1.25 | 12.17 ± 1.04 | 14.67 ± 0.58 | 17.67 ± 0.58 |
| *P. glabrum* CCM F-310 | pos$^{++}$ | 12.00 ± 0.00 | 16.67 ± 0.58 | 20.33 ± 0.58 | 23.67 ± 0.58 |
| *F. graminerum* CCM F-683 | pos$^+$ | 12.00 ± 0.00 | 24.67 ± 0.58 | 25.67 ± 0.58 | 27.67 ± 1.15 |
| | SDA (a$_w$ = 0.94) Ø (mm) | | | | |
| *C. cladosporioides* CCM F-348 | pos$^+$ | 8.67 ± 0.58 | 12.33 ± 1.15 | 12.33 ± 1.15 | 13.00 ± 1.00 |
| *C. herbarum* CCM F-455 | neg | pos+ | 7.67 ± 0.58 | 7.67 ± 0.58 | 10.00 ± 0.00 |
| *P. chrysogenum* CCM F-8322 | pos$^+$ | 14.67 ± 0.58 | 20.33 ± 0.58 | 20.33 ± 0.58 | 28.67 ± 0.58 |
| *P. crustosum* CCM F-8322 | pos$^{++}$ | 13.67 ± 0.58 | 19.00 ± 0.00 | 19.00 ± 0.00 | 25.00 ± 0.00 |
| *P. griseofulvum* CCM F-8006 | pos$^{+++}$ | 15.00 ± 0.58 | 19.33 ± 0.58 | 19.33 ± 0.58 | 24.00 ± 0.00 |
| *P. glabrum* CCM F-310 | pos$^{+++}$ | 18.33 ± 0.58 | 27.67 ± 0.58 | 27.67 ± 0.58 | 36.67 ± 1.53 |
| *F. graminerum* CCM F-683 | pos$^+$ | 27.67 ± 1.15 | 43.00 ± 1.00 | 43.00 ± 1.00 | 46.33 ± 1.53 |

h—hours; neg.—no growth; pos$^+$—growth observed at 10× magnification; pos$^{++}$—growth observed with the naked eye on the surface of the disk; pos$^{+++}$—growth observed with the naked eye even around the disk.

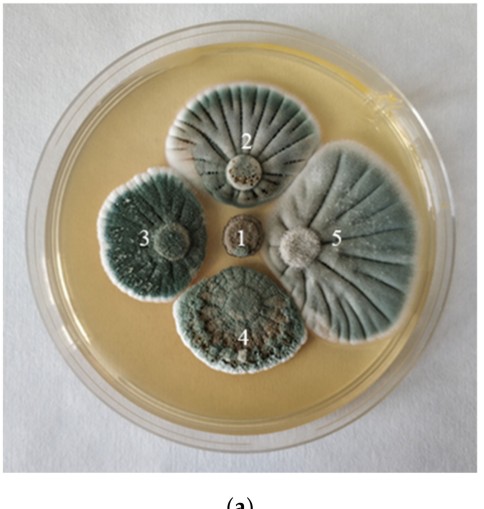 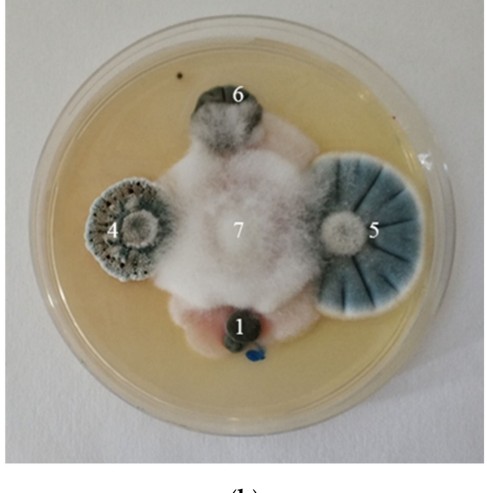

(**a**)                                                                    (**b**)

**Figure 2.** Antifungal activity of mold species tested in this study: (**a**) *C. cladosporioides* CCM F-348 (1), *P. chrysogenum* CCM F-362 (2), *P. crustosum* CCM F-8322 (3), *P. griseofulvum* CCM F-8006 (4), *P. glabrum* CCM F-310 (5); (**b**) *C. cladosporioides* CCM F-348 (1), *P. griseofulvum* CCM F-8006 (4), *P. glabrum* CCM F-310 (5), *F. graminearum* CCM F-683 (6), *C. herbarum* CCM F-455 (7).

### 3.3. Effect of Storage Conditions on Egg Quality Parameters

The measurements of selected quality parameters in experimental groups of table eggs are presented in Table 5. Among the five parameters studied, the impact of the storage period was only confirmed in the case of Haugh units and the color of the yolk ($p < 0.05$). As for egg weight, significant differences were observed on Day 21 between experimental Groups B and C ($p < 0.01$). Differences in Haugh units (HU) between experimental groups A and B were only determined on Day 7 ($p < 0.01$). As follows from the results of this study, the most remarkable changes in egg quality were observed when eggs had been stored at laboratory temperature (Group C). Maximum weigh losses were detected between Day 7 and Day 14 of the experiment with an average value of 3.5%. In group C eggs, Haugh

units decreased continuously within the storage period. At the end of experiment (Day 28), most eggs had to be downgraded to Grade B.

**Table 5.** Egg quality parameters in three experimental groups of table eggs (A–C) during storage (mean ± SD).

| Parameter | Group | Storage Period [Days] | | | | ANOVA [*p* Values] | |
| | | 7 | 14 | 21 | 28 | Impact of Storage Conditions | Impact of Storage Period |
|---|---|---|---|---|---|---|---|
| Egg weight [g] | A | 62.47 ± 3.07 [a1] | 60.35 ± 3.55 [a1] | 60.37 ± 2.37 [a12] | 62.80 ± 1.44 [a1] | | |
| | B | 62.25 ± 1.55 [a1] | 62.05 ± 0.85 [a1] | 62.00 ± 0.53 [a1] | 62.43 ± 3.29 [a1] | *p* < 0.01 | *p* > 0.05 |
| | C | 62.43 ± 0.61 [a1] | 56.80 ± 4.20 [a1] | 57.00 ± 1.60 [a2] | 59.15 ± 1.55 [a1] | | |
| Eggshell firmness [kgf] | A | 4.33 ± 0.18 [a1] | 4.67 ± 0.65 [a1] | 4.72 ± 0.29 [a1] | 4.99 ± 0.36 [a1] | | |
| | B | 3.99 ± 2.16 [a1] | 4.70 ± 0.38 [a1] | 2.60 ± 2.49 [a1] | 5.01 ± 0.43 [a1] | *p* > 0.05 | *p* > 0.05 |
| | C | 5.10 ± 0.22 [a1] | 4.08 ± 2.40 [a1] | 5.13 ± 0.33 [a1] | 5.83 ± 0.96 [a1] | | |
| Haugh units [HU] | A | 86.87 ± 2.78 [a1] | 83.65 ± 5.35 [a1] | 78.57 ± 6.49 [a1] | 78.27 ± 7.51 [a1] | | |
| | B | 70.05 ± 5.05 [a2] | 84.40 ± 2.90 [a1] | 69.93 ± 8.56 [a1] | 70.77 ± 12.64 [a1] | *p* < 0.01 | *p* < 0.05 |
| | C | 78.10 ± 4.65 [a12] | 74.85 ± 3.15 [a1] | 70.90 ± 2.40 [ab1] | 66.75 ± 0.25 [b1] | | |
| Yolk color | A | 10.00 ± 0.00 [a1] | 7.00 ± 3.00 [a1] | 9.00 ± 1.00 [a1] | 7.00 ± 2.65 [a1] | | |
| | B | 10.00 ± 0.00 [a1] | 10.33 ± 0.58 [a1] | 7.67 ± 2.31 [a1] | 9.67 ± 1.15 [a1] | *p* > 0.05 | *p* < 0.05 |
| | C | 10.00 ± 0.00 [a1] | 9.33 ± 0.58 [a1] | 9.00 ± 1.00 [ab1] | 6.67 ± 1.53 [b1] | | |
| Grade of quality | A | AA [a1] | AA [a1] | AA [a1] | A–AA [a1] | | |
| | B | A–AA [a1] | A–AA [a1] | A–AA [a1] | B–AA [a1] | *p* > 0.05 | *p* > 0.05 |
| | C | AA [a1] | A–AA [a1] | B–AA [a1] | B–A [a1] | | |

Different superscripts in the same row ([a, b]) or column ([1,2]) indicate that mean values differ significantly (Tukey's test, *p* < 0.05).

### 3.4. Statistical Analysis

Statistical analysis of results achieved in this study is shown in Table 6. The Pearson's correlation matrix demonstrates reciprocal correlations between individual quality parameters studied in experimental groups of table eggs. Significant differences are highlighted in bold. Correlations between the following parameters at the level of $p < 0.05$ can be seen in this table: eggshell firmness and the count of molds on the shell surface (r = −0.385), Haugh units and quality grades of table eggs (r = 0.717), Haugh units and relative humidity (r = 0.440), yolk color and the count of molds present in the yolk (r = −0.345), egg weight and storage temperature (r = −0.490), egg weight and relative humidity (r = 0.334), the count of molds in the egg white and in the egg yolk (r = 0.748), grade of quality and relative humidity (r = 0.351), storage temperature and relative humidity (r = −0.402), storage temperature and dew point temperature (r = 0.816).

**Table 6.** Pearson's correlation matrix for egg quality parameters (significant differences are highlighted in bold).

| Parameter | Shell Firmness | Haugh Units | Yolk Color | Egg Weight | CFU/* Eggshell * | CFU/g Albumen * | CFU/g Yolk * | Grade of Quality | Temperature | Relative Humidity | Dew Point |
|---|---|---|---|---|---|---|---|---|---|---|---|
| Shell firmness | | 0.188 | 0.078 | −0.101 | **−0.385** | −0.204 | −0.242 | 0.074 | 0.267 | −0.136 | 0.199 |
| Haugh units | 0.188 | | 0.206 | 0.109 | 0.108 | 0.054 | −0.031 | **0.717** | −0.079 | **0.440** | 0.205 |
| Yolk color | 0.078 | 0.206 | | 0.215 | 0.046 | −0.252 | **−0.345** | 0.233 | −0.154 | −0.155 | −0.249 |
| Egg weight | −0.101 | 0.109 | 0.215 | | −0.146 | −0.148 | −0.239 | 0.136 | **−0.490** | 0.334 | −0.308 |
| CFU/eggshell * | −0.385 | 0.108 | 0.046 | −0.146 | | 0.031 | 0.209 | 0.073 | −0.130 | −0.030 | −0.147 |
| CFU/g albumen * | −0.204 | 0.054 | −0.252 | −0.148 | 0.031 | | 0.748 | 0.148 | −0.038 | 0.148 | 0.067 |
| CFU/g yolk * | −0.242 | −0.031 | −0.345 | −0.239 | 0.209 | 0.748 | | −0.075 | −0.011 | 0.054 | 0.009 |
| Grade of quality | 0.074 | 0.717 | 0.233 | 0.136 | 0.073 | 0.148 | −0.075 | | −0.135 | **0.351** | 0.084 |
| Temperature | 0.267 | −0.079 | −0.154 | −0.490 | −0.130 | −0.038 | −0.011 | −0.135 | | **−0.402** | **0.816** |
| Relative humidity | −0.136 | 0.440 | −0.155 | 0.334 | −0.030 | 0.148 | 0.054 | 0.351 | −0.402 | | 0.189 |
| Dew point | 0.199 | 0.205 | −0.249 | −0.308 | −0.147 | 0.067 | 0.009 | 0.084 | 0.816 | 0.189 | |

CFU colony forming unit. * number of molds.

The MFA statistical method was applied to the data of microbial and quality parameters of experimental eggs and parameters of storage conditions, whereas storage conditions and storage period were set as the main qualitative factors. The results of MFA show five selected components, which explain 77.78% of the total variation in the dataset. The first dimension (Dim1) explains 20.78%, dimension 2 (Dim2) 18.03%, dimension 3 (Dim3) 16.05%, dimension 4 (Dim4) 12.54% and dimension 5 (Dim5) 9.34% of variation.

The contribution of the data analysed in Dim1 was related to parameters of storage conditions (39.32%, r = 0.95). The first two dimensions explained a total of 38.81% of variance (see Figure 3). The highest contribution in Dim1 included relative humidity (r = 0.51), dew point temperature (r = −0.70) and storage temperature (r = −0.95). Dim2 was characterized by the contribution of the effect of storage conditions (43.40%, r = 0.92) on the analyzed parameters.

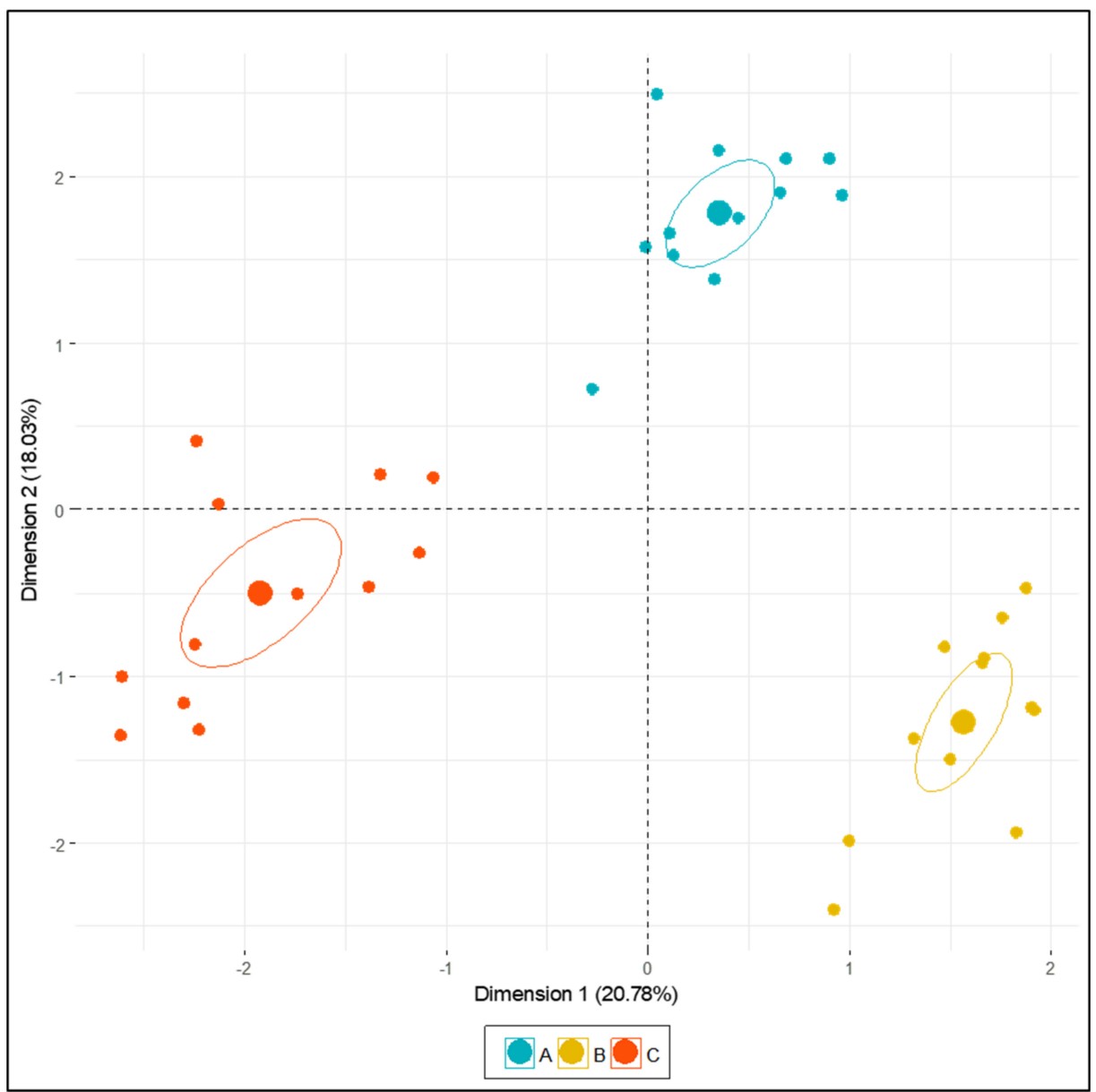

**Figure 3.** Plot of individuals in the first and second extracted dimension under different storage conditions. A—table eggs stored at average temperature of (11.43 ± 0.20) °C, relative humidity of (79.00 ± 2.94)% and the dew point of (7.90 ± 0.47) °C. B—table eggs stored at average temperature of (3.33 ± 1.1) °C, relative humidity of (57.76 ± 5.30)% and the dew point of (−4.25 ± 1.96) °C. C—table eggs stored at average temperature of (20.60 ± 1.04) °C, relative humidity of (43.55 ± 5.93)% and the dew point of (7.60 ± 2.39) °C.

Parameters in the first two dimension were correlated on statistical significant level $\alpha < 0.05$ (see Figure 4). Dim3 was related mostly to the microbial parameters of experimental groups of eggs (44.35%, r = 0.89). Correlation coefficients for microbial parameters in Dim3 were determined as follows: CFU in the egg yolk (r = 0.77), CFU in the egg white (r = 0.71) and CFU in the eggshell (r = 0.54). Dim4 shows relations mainly in quality parameters of experimental egg groups (23.06%, r = 0.66). In Dim4, there were statistically significant correlations for yolk color (r = 0.61) and egg quality grade (r = 0.37). Dim5 explains mostly the effect of storage period on monitored egg parameters (88.53%, r = 0.98).

From the results obtained by MFA, it follows that the effects of storage conditions on experimental eggs were statistically significant. The MFA method showed differences between experimental egg groups A, B and C, which were visualized in different segments of individual plots (see Figures 3 and 4).

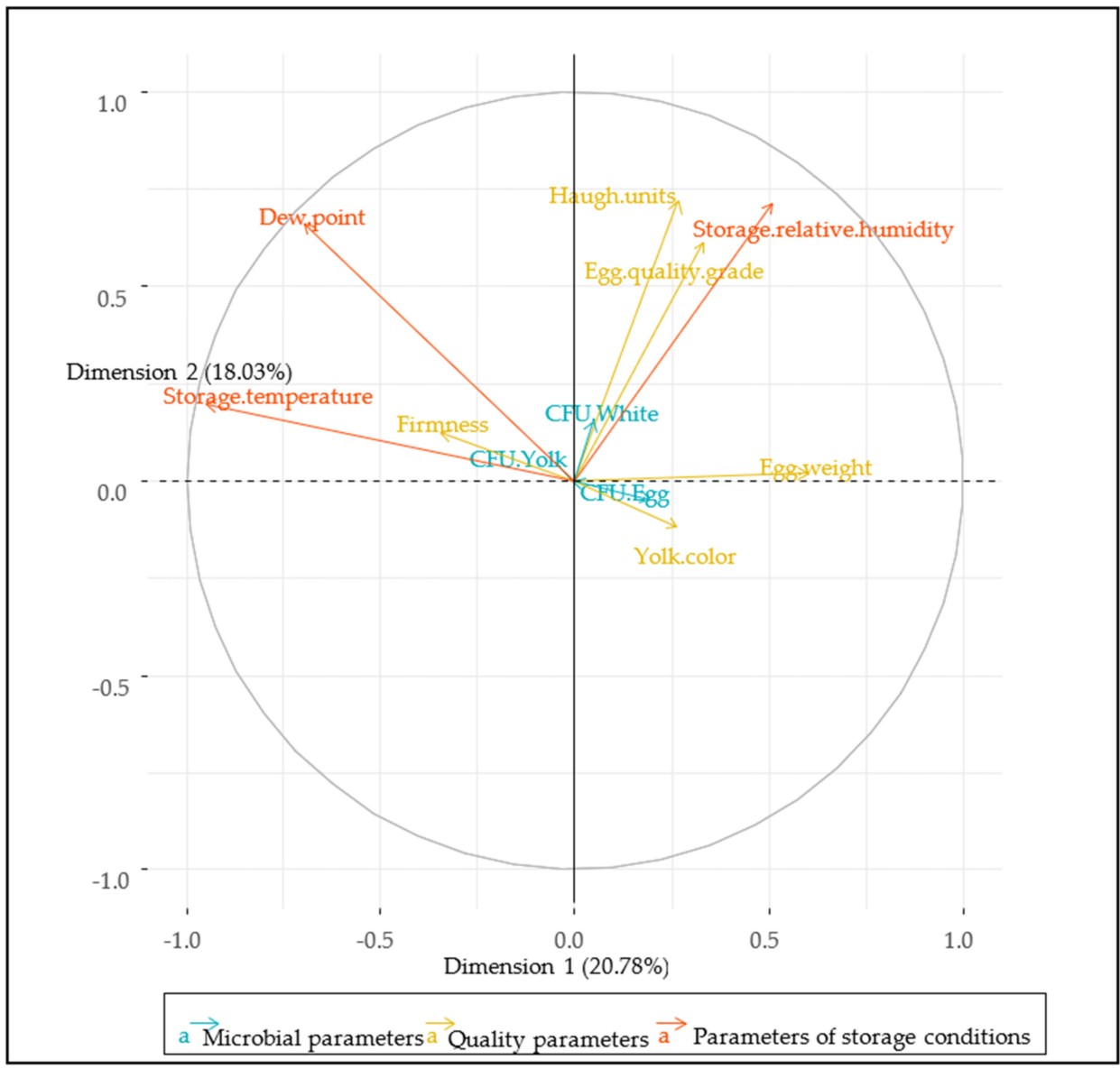

**Figure 4.** Correlation plot of variables in the first and second extracted dimension under different storage conditions.

## 4. Discussion

It is a well-known fact that the egg quality and the growth of contaminating mycoflora on the shell surface depend mainly on storage conditions. The aim of the present study

was to examine the interaction between the storage time, *C. cladosporioides* contamination and their effect on the quality of table eggs.

Results of previous studies performed at the Department of Food Hygiene, Technology and Safety of the University of Veterinary Medicine and Pharmacy in Košice have demonstrated that the genus *Cladosporium* is one of the most common type of mold on the eggshells in egg storage facilities [34,35]. Due to tolerance to low temperatures, *C. cladosporioides* is able to colonize refrigerated food products as well as refrigerator interiors [36]. In this study, egg storage temperatures were chosen in accordance with Commission Regulation No 589/2008 [37], with the minimum limited to 5 °C. As the storage temperatures ensure an optimum egg quality within the 28 days, most egg producers recommend storing the eggs at temperatures between 5 °C and 18 °C.

The results of this study demonstrate that the average counts of molds on/in the eggs remained practically unchanged across the entire storage period. However, noticeable changes were observed in the composition of mycoflora during egg storage. Apart from *C. cladosporioides*, the fungal species used for initial eggshell contamination, the presence of another two mold genera (*Penicillium* spp. and *Fusarium* spp.) as well as yeasts have also been detected in the experimental eggs.

Fluctuating air humidity results in a transient water supply affecting both spore germination and fungal growth. Wu and Wong [38] reported growth retardation of *C. cladosporioides* spores exposed to daily cycles with various combinations of wet and dry periods. The longer the phase of growth retardation, the higher the intracellular concentration of hydrogen peroxide. Thus, the fluctuating air humidity may result in slowing growth of *C. cladosporioides* and subsequent oxidative stress.

Visible growth of either *C. cladosporioides* or the other mold genera was not detected on/in experimental table eggs across the entire period of this study. This result could probably be explained by the variable germination ability of spores, which differs considerably depending on environmental conditions [39] and low relative humidity during egg storage. As for *C. cladosporioides*, the minimum values of available water ($a_w$) required for spore germination range between 0.85 and 0.91 and those for sporulation between 0.87 and 0.95 [20]. Mold development on the shell surface can also be facilitated by low temperatures, provided that the relative humidity is higher than 90% [19]. The spores of *Penicillium* spp. germinate from 0.79 $a_w$ (*P. piceum*) to 0.83 $a_w$ (*P. roqueforti*) [40].

The isolates of *Penicillium* spp. were dominant on experimental days when the maximum relative humidity and dew point temperature were observed. The proportion of penicillia on these days varied from 68 to 91% depending on the particular experimental group. The results of this study indicate that *Penicillium* and *Fusarium* species grow faster at higher $a_w$ (0.91–0.94). Thus, they may pose a risk to the consumer in the case of eggshell contamination. While the adverse effects of *C. cladosporioides* metabolites are still under investigation [41], the toxicity of secondary metabolites produced by *Penicillium* and *Fusarium* species has already been reliably confirmed [42–44]. Among the hundreds of identified mycotoxins endangering human or animal health, aflatoxins, ochratoxin A, patulin and fusarial mycotoxins (fumonisins, zearalenon and nivalenol/deoxynivalenol) are of greatest concern [45]. *P. verrucosum* and *P. nordicum* have been recognized as major producers of ochratoxin A [46]. The presence of *P. verrucosum* was also detected among fungal contaminants of table eggs [47], indicating a possible risk of egg contamination with ochratoxin A. *Fusarium* sp. is reported to produce a number of mycotoxins, including trichothecens such as deoxynivalenol (DON), nivalenol (NIV), T-2 and HT-2 toxins, as well as zearalenon (ZEN) and fumonisins [48]. As reported by Tomczyk et al. [49], contamination of table eggs with *F. culmorum* has resulted in the presence of DON, 3-AcDON and NIV in the egg contents. Growth of *Aspergillus flavus* on the eggshell may lead to contamination of egg contents with aflatoxins [50].

The genus *Penicillium* is quite common in table eggs [47]. The results of a Nigerian study confirmed the presence of this fungus in 82.5% of sampled eggs [16]. The combi-

nation of appropriate environmental conditions and a poor level of hygiene may lead to proliferation of microorganisms including molds [18,49,51].

The growth of competetive mycoflora can also be affected by mutual interactions [52]. In this study, antifungal activity of *C. cladosporioides* against *P. chrysogenum*, *P. crustosum* and *P. griseofulvum*, as well as mutual inhibition among *Penicillium* species tested, has been confirmed. *C. cladosporioides* belongs to xerophilic and psychrophilic molds which are able to produce at least three metabolites (cladosporin, isocladosporin and 5′-hydroxyasperentin) and their derivates, showing antifungal activity against pathogenic mold genera including *Aspergillus* spp., *Penicillium* spp., *Phomopsis viticola*, *Colletotrichum acutatum*, *Colletotrichum fragariae*, *Colletotrichum gloeosporioides* [41,53].

The *Penicillium* genus is known to produce bioactive molecules with a strong antifungal effect [54–57]. Mutual antifungal activity against at least one of the toxigenic *P. echinulatum* and *P. commune* strains was already confirmed in 85 strains of 17 *Penicillium* species, including *P. chrysogenum P. crustosum*, and *P. griseofulvum* [57].

As follows from the results of this study, storage conditions can significantly influence egg quality parameters. Based on Haugh units, experimental eggs were classified into three quality grades (AA, A and B). In extra fresh eggs (AA grade), the HU were above 72. Numerous factors are reported to influence HU quality grades, and among them, storage temperature and storage period are of greatest concern [58]. In this study, a continuous decrease in both HU and yolk color was observed during the egg storage period in all the three experimental groups ($p < 0.05$). On Day 7, the differences between experimental groups A and B were significant ($p < 0.01$). As for egg weight, noticeable changes were confirmed on Day 21 between experimental groups B and C ($p < 0.01$). The most significant decrease in freshness was observed in group C experimental eggs stored at temperatures between (20.00 ± 0.86) °C and (22.09 ± 0.13) °C and air humidity between (40.75 ± 3.42)% and (45.47 ± 1.49)%. At the end of the storage period (Day 28), the following egg quality parameters were observed in Group C: egg weight (59.15 ± 1.55) g, HU (66.75 ± 0.25), yolk color (6.67 ± 1.53) and quality grade (A–B). Other studies reported similar effects of storage period and storage temperature on the egg quality parameters [50,55–63]. Prolonged storage period has resulted in a gradual decrease in the egg weight, HU and yolk color, this being less significant at low storage temperatures (0–5 °C) and more noticeable at higher temperatures (20–29 °C).

Fedde et al. [59] found out that the quality parameters of eggs stored for 9 weeks in the cold were simiar to those observed for eggs stored at room temperature for 3 weeks. Refrigeration storage at (4 ± 2) °C is able to maintain safety and nutritive value of table eggs for approximately 4–5 weeks [60–63]. A decrease in HU from 91.4 to 76.3 was reported for eggs stored at 5 °C for 10 days [64]. An increase in storage temperature to 29 °C has lead to significant weight losses (1.74% within 5 days and 3.67% within 10 days) accompanied with a decrease in HU from 87.62 to 60.92 [65]. Similar results were also obtained in other studies [66,67]. The decrease in HU may result from the escape of carbon dioxide from the egg white, leading to a watery consistency and lower height of the thick albumen—the two parameters considered in HU calculation [68].

Changes in yolk color result from aging of the yolk membrane. Water from the egg albumen is absorbed by the yolk, where it dilutes the yolk pigments and makes the color paler. Long-term egg storage may also result in the penetration of albumen proteins into the egg yolk, causing a decrease in yolk color as well [69]. Elevated storage temperatures accelerate the destruction of the protein structure in both the thick albumen and the vitelline membrane [70]. All the above-mentioned changes have been observed even after a 2-day storage of table eggs at 29 °C [65].

Loss in egg weight is more commonly associated with water evaporation than with the escape of carbon dioxide ($CO_2$), ammonia, nitrogen and hydrogen sulphide through the shell pores [64]. More significant weight losses are reported for eggs stored at higher ambient temperature and lower relative humidity [71]. In this study, the most noticeable weight losses were observed in Group C experimental eggs stored for 28 days at the highest

temperature (20 °C) and lowest relative humidity (45%). Moreover, the results of this study revealed a close reciprocal correlation between relative air humidity and eggshell firmness, where firmness was significantly reduced with an increase in air humidity in the egg storage room. As reported by Tyler and Geake [72], moisture in the air makes the eggshell weaker. However, the shell regains its firmness after drying in air.

## 5. Conclusions

Despite of initial eggshell contamination with spore suspension of *C. cladosporioides*, the results of this study did not confirm any visible growth of molds on/in table eggs stored for 28 days at three different temperatures (3 °C, 11 °C and 20 °C) and relative humidity between 45 to 80%. As demonstrated in this study, high relative air humidity in the egg storage room reduced the growth of *C. cladosporioides*, probably due to oxidative stress resulting from the previous fluctuating humidity values, and favored the growth of *Penicillium* spp.

Comparison of growth characteristics among seven fungal species (*C. cladosporioides* CCM F-348, *C. herbarum* CCM F-455, *P. chrysogenum* CCM F-362, *P. crustosum* CCM F-8322, *P. griseofulvum* CCM F-8006, *P. glabrum* CCM F-310, *F. graminearum* CCM F-683) on the selective culture media (DRBC and SDA) has demonstrated that both *Cladosporium* species grow 24–72 h later than *Penicillium* and *Fusarium* species on the eggshell surface. Furthermore, the results of this study confirmed the inhibition activity of *C. cladosporiorides* CCM F-348 against four *Penicillium* species tested but not against *F. graminerum* CCM F-683. Both *Penicillium* and *Fusarium* species showed more intense growth with larger colony diameters on the surface of the eggshell at high values of relative air humidity, thus increasing the risk to consumers due to possible contamination of egg contents with mycotoxins.

Based on the results obtained, it can be concluded that the most appropriate temperature for egg storage is 11 °C, and the optimum range of relative humidity lies between 75% and 81%. Table eggs stored under such conditions maintained their grade of quality within the entire storage period of 28 days. The other quality parameters also remained unaffected, except for yolk color, which declined steadily during egg storage. In terms of egg quality, the least appropriate conditions for egg storage are temperatures close to 20 °C. Due to weight loss, most eggs stored at this temperature had to be downgraded at the end of the shelf life period. In addition, increased numbers of fungal spores on the egg surface adversely affected the firmness of the eggshell as well as color of the yolk. The results of this study have demonstrated that storage conditions are crucial for maintaining egg quality and mycological safety of table eggs for consumers.

**Author Contributions:** P.J. and M.P.; methodology, P.J.; software, B.S.; formal analysis, M.P.; investigation, S.D. and I.R.; data curation, B.S.; Writing—Original draft preparation, P.J.; Writing—Review and editing, M.P.; visualization, I.K.; supervision, J.N.; project administration, P.J.; funding acquisition, S.M. All authors have read and agreed to the published version of the manuscript.

**Funding:** This research was funded by the Operational Program Integrated Infrastructure within the project: Demand-driven research for the sustainable and inovative food, Drive4SIFood 313011V336, cofinanced by the European Regional Development Fund (70%) and by the Cultural and Educational Grant Agency of the Ministry of Education, Science, Research and Sport of the Slovak Republic and the Slovak Academy of Sciences KEGA 013UVLF-4/2021(30%).

**Institutional Review Board Statement:** Not applicable.

**Informed Consent Statement:** Not applicable.

**Data Availability Statement:** The data presented in this study are available on request from the corresponding author.

**Conflicts of Interest:** The authors declare no conflict of interest.

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
