# Peer review of "Effect of Cladosporium cladosporioides on the Composition of Mycoflora and the Quality Parameters of Table Eggs during Storage"

_processes, doi:10.3390/pr9040613_

Round 1

Reviewer 1 Report

Article entitled „Effect of Cladosporium cladosporioides on composition of mycoflora and quality parameters of table eggs during storage” presents well-thought-out, well-planned and properly conducted research. The authors wrote a crude introduction justifying the research, thoroughly described the methodology and professionally presented the results. The conclusions of the research relate to the application of the results in practice.

I only have a few minor comments to the authors:

- page 1 line 29 - Dot after (not before) quoting

- please use consistently one of the terms occuring in the text: filamentous fungi/ micromycestes/ molds/microscopic filamentous fungi

- page 2 line 77 - lack of space

- page 4 line 173 section title „Comparison of growth among Cladosporium, Penicillium and Fusarium species” – in my opinion better will be „Characteristics of growth and interreactions of selected mould species”

- page 12 line 361 „Department of Food hygiene andTechnology of the University of Veterinary Medicine and Pharmacy in Košice” it shoul be „Department of Food Hygiene..”

- page 13 line 398 the authors wrote „the toxicity of secondary metabolites produced by Penicillium and  Fusarium species has already been reliably confirmed” - whether it concerns general toxinogenicity or the production of toxins on table eggs. Do the authors know if and what mycotoxins can be found in eggs? please add a comment in the text

Reviewer 2 Report

The manuscript by Jevinová et al. deals with conditions used for egg storage and the proliferation of mycoflora. 120 eggs were subdivided in three groups and sampled at regular interval. Molds were isolated and characterized and their succession evaluated. Ihnibitory ineractions among developped molds were also evaluated. The authors conclude that the most appropriate temperature for egg storage (up to 28 days) is 11 °C, with an optimum range of relative humidity laying in the range 75-81%. 

The manuscript is well written, methods are clearly described and results well reported. 

In my opinion the paper should be accepted in its present form.

Author Response

Response to Reviewer 2 Comments

The authors would like to thank Reviewer 2 for reading and approving the submitted manuscript for publication in the Processes MDPI journal.